# Zero-temperature glass transition in two dimensions

Ludovic Berthier [1], Patrick Charbonneau [2,3], Andrea Ninarello [4], Misaki Ozawa[1] & Sho Yaida[5]

Liquids cooled towards the glass transition temperature transform into amorphous solids that have a wide range of applications. While the nature of this transformation is understood rigorously in the mean-field limit of infinite spatial dimensions, the problem remains wide open in physical dimensions. Nontrivial finite-dimensional fluctuations are hard to control analytically, and experiments fail to provide conclusive evidence regarding the nature of the glass transition. Here, we develop Monte Carlo methods for two-dimensional glass-forming liquids that allow us to access equilibrium states at sufficiently low temperatures to directly probe the glass transition in a regime inaccessible to experiments. We find that the liquid state terminates at a thermodynamic glass transition which occurs at zero temperature and is associated with an entropy crisis and a diverging static correlation length. Our results thus demonstrate that a thermodynamic glass transition can occur in finite dimensional glass-formers.

[1] Laboratoire Charles Coulomb (L2C), University of Montpellier, CNRS, F-34095 Montpellier, France. [2] Department of Chemistry, Duke University, Durham, NC 27708, USA. [3] Department of Physics, Duke University, Durham, NC 27708, USA. [4] CNR-ISC, UOS Sapienza, Piazzale A. Moro 2, IT-00185 Roma, Italy. [5] Facebook AI Research, Facebook Inc., Menlo Park, CA 94025, USA. Correspondence and requests for materials should be addressed to L.B. (email: ludovic. berthier@umontpellier.fr)

Difficult scientific problems can drastically simplify in some unphysical limits. For instance, very large dimensions ($d \to \infty$, where $d$ is the spatial dimensions) give relevant fluctuations a simple mean-field character[1], and one-dimensional ($d = 1$) models can often be treated exactly. Yet these two solvable limits are crude idealizations of our three-dimensional reality. The rich theoretical arsenal developed to interpolate between them has revealed the highly nontrivial role of spatial fluctuations in all areas of science. In particular, as the number of spatial dimensions decreases, a phase transition may change nature or even disappear. Dimensionality thus provides a key tool for understanding the essence of many natural phenomena.

The glass transition from a viscous liquid to an amorphous solid is no exception[2]. Its mean-field description, which becomes mathematically exact as $d \to \infty$, explains the dramatic slowdown of glass-forming liquids through the rarefaction of the number of glassy metastable states upon approaching a critical temperature, $T_K$[3,4]. The configurational entropy, $s_{\mathrm{conf}}$, which is the logarithm of the number of such states, becomes subextensive when $T \le T_K$. The equilibrium glass transition thus corresponds to an entropy crisis, a hypothesis first suggested by Kauzmann in his visionary analysis of experimental data[5] and initially formalized by Gibbs and DiMarzio[6] in the context of a lattice polymer model.

The broad discussion that has since ensued[2] has notably tried to describe the role of finite-$d$ fluctuations beyond the mean-field framework[7–12], relating in particular the vanishing of $s_{\mathrm{conf}}$ to a diverging point-to-set correlation length, the key quantity for characterizing nonperturbative fluctuations in glass formers[13]. These fluctuations, however, make it difficult to examine finite-dimensional glass formers analytically, even for simple models composed of point-particles such as those we study here. Exploring a broader diversity of models, from polymer[14] to anisotropic patchy[15] models, may yet provide additional theoretical insight.

Meanwhile, Kauzmann's intuition has been repeatedly validated by experiments[16,17], but the conceptual and technical limits of his results have not been lifted. Current experiments access essentially the same restricted temperature range as his 70-year old work. Theory and experiments thus currently fail to assess the status of the Kauzmann transition in finite $d$, or whether new mechanisms qualitatively change the underlying physics[18,19]. Experimentally, it thus remains controversial whether the trend discovered by Kauzmann survives at much lower temperatures; entropy could go smoothly to zero[20,21], or to a finite residual value as temperature vanishes[15,22,23].

In this context, computer simulations are especially valuable. They allow direct measurements of both the configurational entropy and the point-to-set correlation length for realistic models of finite-dimensional glass formers[2]. The recent development of the swap Monte Carlo algorithm (SWAP) further allows the exploration of a temperature regime that experiments cannot easily access[24], even using ultrastable glassy materials[25]. This has consolidated and extended Kauzmann's experimental findings for three-dimensional glass formers[26]. Here, we report that SWAP is so efficient in $d = 2$ that it provides access to a temperature regime equivalent to experimental timescales $10^{18}$ larger than the age of the universe. This remarkable advance gives very strong evidence of a thermodynamic glass transition at $T_K = 0$ for $d = 2$, accompanied by an entropy crisis and the divergence of the point-to-set correlation length. Our results thus illuminate the low-dimensional fate of the glass transition and shed light on the nature of glassy dynamics in $d = 2$[27–30].

## Results

**Model and macroscopic behavior.** We study a two-dimensional mixture of soft particles interacting with a $1/r^{12}$ purely repulsive power-law pair potential and a size polydispersity chosen to minimize demixing, fractionation, and crystallization (see Methods). The average particle diameter is used as unit length, and the strength of the interaction potential as unit temperature. SWAP is implemented following the methodology recently validated for $d = 3$[24]. Systems ranging from $N = 300$ to $N = 20{,}000$ particles within a periodic box are used to carefully track finite-size effects in both dynamics and thermodynamics. We mainly present results of $N = 1000$. Whereas experimental systems are typically composed of more complex particles (such as large molecules or polymers), the exact mean-field theory has thus far only been developed for the same type of point particles as we simulate here. In addition, such models have become a standard to study fundamental aspects of the glass transition, and are good representations of colloidal glasses.

Figure 1a shows that the static structure factor $S(k)$ evolves smoothly over a broad temperature range, from the onset temperature $T_{\mathrm{onset}} = 0.250$ down to $T = 0.026$, which is the lowest temperature for which our strict equilibrium criteria are met. The typical low-temperature configuration depicted in Fig. 1b shows that particles of different sizes are well mixed, and that local ordering is extremely weak. In fact, no crystallization event was ever observed in our simulations, and the correlation lengths extracted from the pair correlation function

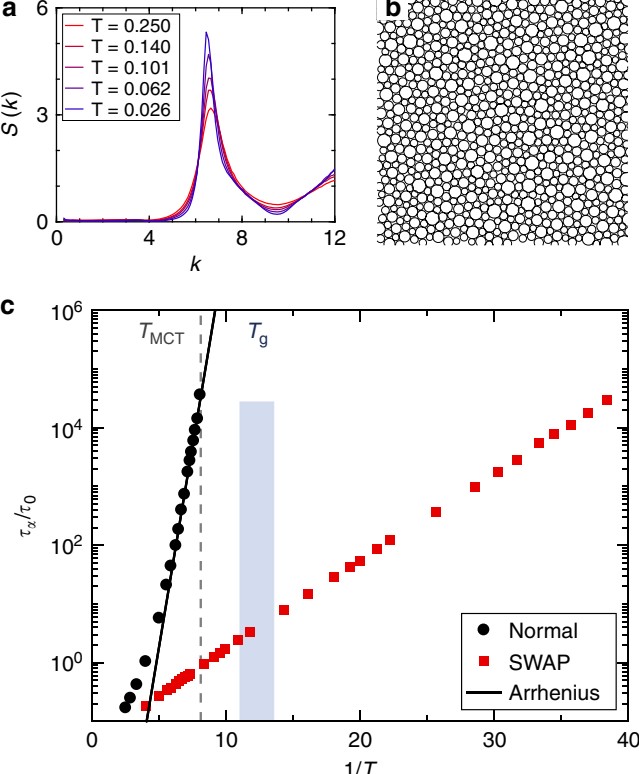

**Fig. 1** Statics and dynamics of the $d = 2$ glass former. **a** The smooth evolution of the static structure factor from $T_{\mathrm{onset}}$ down to the lowest studied temperature $T = 0.026$ indicates that the system remains fully amorphous at all $T$. **b** Snapshot of an equilibrium configuration at $T = 0.026$. **c** Arrhenius representation of the structural relaxation time $\tau_\alpha$ using SWAP and normal Monte Carlo dynamics, rescaled by the relaxation time at the onset temperature. The mode-coupling temperature, $T_{\mathrm{MCT}}$ (gray dashed line), and the estimated range of experimental glass temperature, $T_g$ (navy strip), are indicated. The Arrhenius fit to the low-$T$ data provides a lower bound for the growth of $\tau_\alpha$. SWAP can equilibrate systems down to $T \approx 0.3T_g$, where the Arrhenius fit gives $\tau_\alpha^{\mathrm{normal}}/\tau_0 \sim 10^{46}$

for translational and bond-orientational orders evolve modestly with $T$ (see Supplementary Note 1). In other words, the model is an excellent glass former.

The bulk dynamics and equilibration are captured by the bond-orientational order time correlation, $C_\psi(t)$, which is not affected by long-time tails observed in simple two-dimensional fluids[27,31]. The $1/e$ decay of $C_\psi(t)$ robustly defines bulk relaxation timescales $\tau_\alpha$ both for SWAP ($\tau_\alpha^{\mathrm{SWAP}}$) and normal ($\tau_\alpha^{\mathrm{normal}}$) Monte Carlo dynamics (Fig. 1c). We normalize these timescales by $\tau_0 \equiv \tau_\alpha^{\mathrm{normal}}(T_{\mathrm{onset}})$. In agreement with earlier works[27], we find that translational correlation functions suffer from large finite-size effects, but that subtracting long-range Mermin–Wagner translational fluctuations results in system-size independent measurements[28–30] consistent with bond-orientational dynamics (see Supplementary Fig. 1). The normal dynamics exhibits a well-known super-Arrhenius growth of $\tau_\alpha$. Fitting its temperature evolution to a power-law divergence situates the mode-coupling crossover at $T_{\mathrm{MCT}} = 0.123$, which is roughly the lowest temperature accessible with this dynamics. Earlier work showed that thermalization can be achieved below $T_{\mathrm{MCT}}$ using Monte Carlo simulations[32]. Following ref. [24], we estimate the narrow range within which the experimental glass temperature takes place as $T_g \in [0.0738, 0.0907]$. (Henceforth we set $T_g = 0.082$.) The lower end of this interval stems from an Arrhenius fit which provides a lower bound to the true $\tau_\alpha$. By all estimates, SWAP dynamics is clearly much faster than the normal one. The speedup is about 5 orders of magnitude at $T_{\mathrm{MCT}}$, 10 at $T_g$, and the Arrhenius lower bound suggests a formidable 42 order-of-magnitude speedup at $T = 0.026$. Using an atomistic value, $\tau_0 = 10^{-10}$ s, converts this estimate to $\tau_\alpha = 10^{36}$ s, or approximately $10^{18}$ times the age of the universe. Such a "cosmological" speedup leaves no doubt that the SWAP equilibration algorithm largely bypasses the slowdown associated with the glass transition in $d = 2$.

**Configurational entropy.** This computational advance permits the study of the $d = 2$ configurational entropy and its relationship to the putative entropy crisis far beyond the previous work[33]. Extending earlier work on $d = 3$ systems[26], we obtain independent estimates of $s_{\mathrm{conf}}$ using state-of-the-art methodologies, see Fig. 2a. Technical details are described in Supplementary Note 2. The first estimate stems from subtracting the vibrational contribution, measured by minimizing the potential energy of the system to an inherent structure and obtaining its vibrational spectrum, from the total liquid entropy[34]. This potential energy landscape (PEL) approach needs to be complemented, for polydisperse systems, with an independent measure of the mixing entropy[35]. Because minor but systematic additional adjustments are then required, two sets of PEL estimates are reported in Fig. 2a. The two are quantitatively close and similarly decrease with $T$, which confirms that methodological details do not affect our results in any essential way. This approach extends $s_{\mathrm{conf}}$ measurements from $1.5T_g$ in earlier $d = 2$ simulations[33] down to a temperature five times smaller, $0.3T_g$.

Our second estimate directly measures the glass entropy by performing a thermodynamic integration from the well-controlled harmonic solid limit. This approach, which is inspired by the Frenkel–Ladd method for crystals[36], was recently adapted to polydisperse amorphous solids[37]. Because it does not count the number of inherent structures but measures instead the entropy of constrained glassy states, it is also very close in spirit (although not equivalent[37]) to the free-energy measurement that makes use of the Franz–Parisi potential[38]. The Frenkel–Ladd estimate is smaller than the PEL ones, as expected, but exhibits a similar temperature dependence.

From the data in Fig. 2a, $s_{\mathrm{conf}}$ seemingly vanishes close to $T_K = 0$. This behavior sharply contrasts with that of three-dimensional glass formers, for which evidence suggests that $T_K > 0$[5,16,17,26]. The impending entropy crisis is expected to give rise to large-scale fluctuations with a growing point-to-set correlation length[13]. We use the computational tools developed in refs. [26,39,40] to analyze the thermodynamic properties of liquids confined within spherical cavities of radius $R$ drawn from a reference equilibrium configuration (see Supplementary Note 3). The distribution $P(Q)$ of the core cavity overlap $Q$ among the confined equilibrium glassy configurations is then analyzed. The point-to-set correlation length, $\xi_{\mathrm{PTS}}$, is determined from the decay with $R$ of the average overlap. This length is then transformed into a third estimate, $s_{\mathrm{conf}} \propto \xi_{\mathrm{PTS}}^{-(d-\theta)}$ with $\theta = 1$. In $d = 2$, this choice of $\theta$ is natural because it both saturates the bound $\theta \leq d - 1$[13] and satisfies the wetting relation $\theta = d/2$[3]. The resulting $s_{\mathrm{conf}}(T) = \xi_{\mathrm{PTS}}(T_g)/\xi_{\mathrm{PTS}}(T)$ in Fig. 2a again has a similar temperature evolution as other estimates.

Figure 2b shows that rescaling all configurational entropies by their value at $T_g$ collapses the entire set of measurements. This robustness is non-trivial because all four estimates make different types of approximations. The agreement of their temperature dependence may thus resolve earlier discrepancies and debates regarding conflicting estimates of the configurational entropy[41,42].

One expects $s_{\mathrm{conf}}$ to vanish linearly, $s_{\mathrm{conf}} \propto (T - T_K)$, but this scaling arguably has a quadratic correction at higher temperatures. We thus perform a quadratic fit to the low-temperature regime, $T < T_g$. This fitting yields $|T_K| \leq 0.003$ for all cases. These estimates of $T_K$ are 10 times smaller than our lowest temperature, $T = 0.026$, and 30 times smaller than $T_g$. The scaling behavior implied by this observation is presented in Supplementary Note 4. Known alternatives to an entropy crisis invoke a change in the concavity of $s_{\mathrm{conf}}$[14,22,43] and should be accompanied by a maximum in the specific heat $c_V$[18,20,21]; we observe neither the convexity (Fig. 2a) nor the specific heat maximum (Fig. 2c). As $T \to T_K$, $c_V$ instead monotonically increases towards a finite value that is larger than the Dulong–Petit law. These observations thus strongly support the occurrence of a non-trivial entropy crisis at $T_K = 0$. The only alternative left is a change of behavior occurring at temperatures even lower than those we can study directly.

**Point-to-set length scale.** The thermodynamic glass transition at $T_K = 0$ also coincides with a divergence of the point-to-set correlation length. We illustrate the physical meaning of this length scale in Fig. 3a in the form of a $(T, 1/R)$ diagram reminiscent of both the Franz–Parisi thermodynamic construction[38] and of the random pinning approach[44,45]. Upon decreasing the cavity size at a given temperature, the system crosses over from a low-$Q$ regime at large $R$ to a high-$Q$ regime at small $R$, as illustrated by the snapshots in Fig. 3a. For any $T > 0$, this crossover around $R \approx \xi_{\mathrm{PTS}}$ corresponds to a finite-size version of the random first-order glass transition with a rarefaction of the number of locally available states as $R$ decreases[46]. The evolution of $P(Q)$ in Fig. 3b indeed exhibits features reminiscent of phase coexistence near an incipient random first-order transition. The crossover also sharpens as $T$ decreases, suggesting that the growing correlation length transforms it into a genuine thermodynamic phase transition as $T \to T_K = 0$. In absolute values, $\xi_{\mathrm{PTS}} \approx 6.5$ at $T = 0.028$, which represents a very large static correlation length for glassy models[26,39,40]. It implies that large clusters comprising about 120 particles are statically correlated, and should thus move collectively to restructure the liquid. These results are consistent with the sharp decay of the configurational entropy in Fig. 2 and the expected dramatic increase of the relaxation time in Fig. 1.

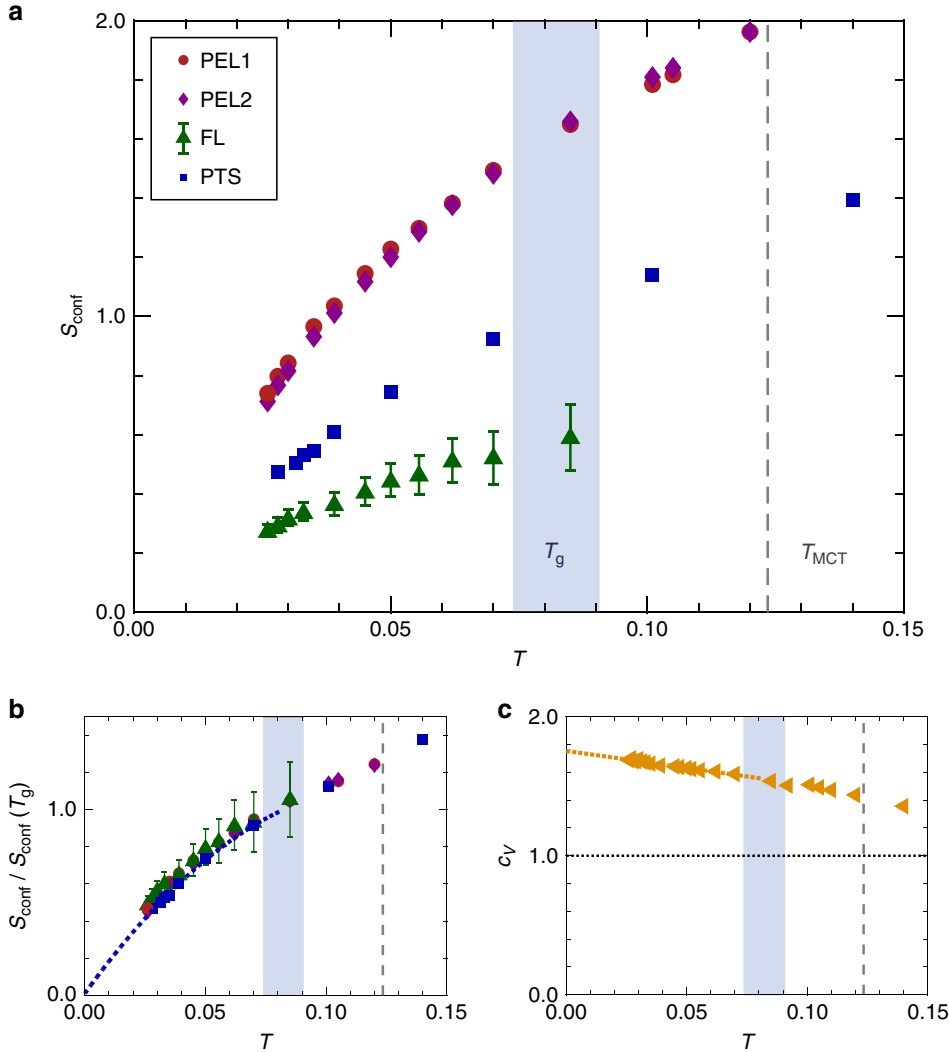

**Fig. 2** Zero-temperature Kauzmann transition. **a** Decrease of the configurational entropy with temperature using the potential energy landscape (PEL), Frenkel–Ladd (FL), and point-to-set (PTS) length estimates. The error bars for FL correspond to the ambiguity of defining the plateau regime in the mean squared displacement of the FL construction. **b** Once rescaled by their value at $T_g$, all estimates evolve nearly identically, and the collapsed data are well fitted by a quadratic function of $T$ for $T < T_g$ (dashed blue line: $s_{conf}(T)/s_{conf}(T_g) = 0.01 + 1.48(T/T_g) - 0.49(T/T_g)^2$ indicates the quadratic fit for the point-to-set estimate). All results are consistent with a linearly vanishing $s_{conf}$ at $T_K = 0$. **c** The specific heat, $c_V$, obtained from the derivative of the potential energy increases monotonically above the Dulong–Petit law for $d = 2$ (dashed horizontal line), which is also consistent with a thermodynamic transition at $T_K = 0$

## Discussion

The problem of the glass transition has two fundamental facets: thermodynamics and dynamics. While the current study focused on the thermodynamics of $d = 2$ glass formers, its dynamical counterpart, which involves obtaining a detailed functional form of the structural relaxation time, remains for now out of reach of computational work. Our results nonetheless suggest that in $d = 2$ the divergence of the relaxation time must take place at zero (rather than at finite) temperature. By identifying the thermodynamic properties that underlie the nature of glassy dynamics in $d = 2$[27–30], our results provide additional evidence that a thermodynamic transition can occur in finite-dimensional systems, and that the lower critical dimension for the long-range amorphous order is $d_L = 2$ (see Supplementary Note 5). This finding lends indirect support to previous observations in $d = 3$[26], and will surely guide future analytical work.

## Methods

**Model**. The glass-forming model we consider consists of particles with purely repulsive soft-sphere interactions, and a continuous size polydispersity. Particle

diameters, $\sigma_i$, are randomly drawn from a distribution of the form: $f(\sigma) = A\sigma^{-3}$, for $\sigma \in [\sigma_{min}, \sigma_{max}]$, where $A$ is a normalization constant. The size polydispersity is quantified by $\delta = \sqrt{\overline{\sigma^2} - \overline{\sigma}^2}/\overline{\sigma}$, where $\overline{\cdots} \equiv \int d\sigma f(\sigma)(\cdots)$, and is here set to $\delta = 0.23$ by imposing $\sigma_{min}/\sigma_{max} = 0.45$. The average diameter, $\overline{\sigma}$, sets the unit of length. The soft-sphere interactions are pairwise and described by an inverse power-law potential

$$v_{ij}(r) = v_0\left(\frac{\sigma_{ij}}{r}\right)^{12} + c_0 + c_1\left(\frac{r}{\sigma_{ij}}\right)^2 + c_2\left(\frac{r}{\sigma_{ij}}\right)^4, \quad (1)$$

$$\sigma_{ij} = \frac{(\sigma_i + \sigma_j)}{2}(1 - \varepsilon|\sigma_i - \sigma_j|), \quad (2)$$

where $v_0$ sets the unit of energy (and temperature with Boltzmann constant $k_B = 1$), and $\varepsilon = 0.2$ quantifies the degree of non-additivity of particle diameters. We introduce $\varepsilon > 0$ to the model in order to suppress fractionation and thus enhance glass form ability[24,47]. The constants, $c_0$, $c_1$, and $c_2$, enforce a vanishing potential and the continuity of the first and second derivatives of the potential at the cut-off distance $r_{cut} = 1.25\sigma_{ij}$. We simulate a system with $N$ particles within a square cell of area $V$ under periodic boundary conditions, at number density $\rho = N/V = 1.01$. Most simulations have $N = 1000$, but systems with $N = 300, 3000, 8000$, and $20,000$ are also studied.

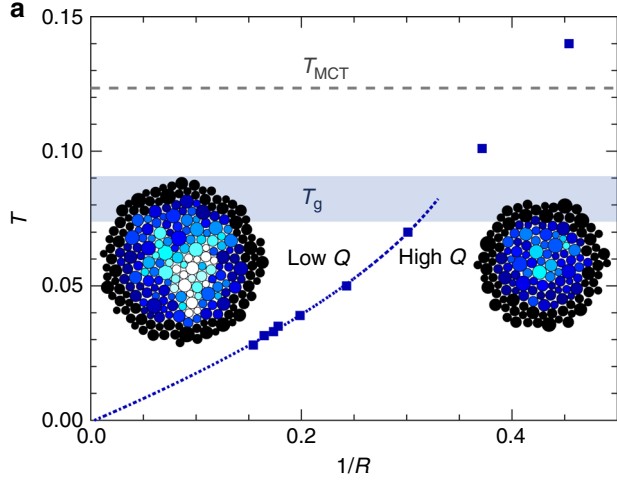

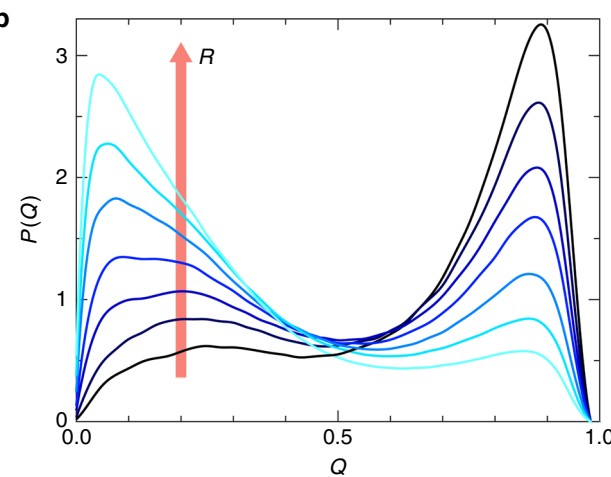

**Fig. 3** Approaching the random first-order transition. **a** Phase diagram showing the low-$Q$ region for large cavities and high-$Q$ region for small cavities, separated by the boundary determined by the point-to-set correlation length, $\xi_{PTS}$. The dashed blue line is the same quadratic fit (after unit conversion) as in Fig. 2b. Inset: Representative configurations with overlap field for $T = 0.035$ at $R = 6.6$ (low $Q$, white) and 4.8 (high $Q$, dark). **b** Evolution of the probability distribution function of overlap $P(Q)$ at $T = 0.035$ from $R = 4.8$ to $R = 6.6$. Bimodality signals a first-order-like phase coexistence

**Observables**. We monitor the system structure with two common liquid state quantities: the pair-distribution function $g(r)$, and the structure factor $S(k) = \langle \rho_{-\mathbf{k}} \rho_{\mathbf{k}} \rangle / N$, where $\rho_{\mathbf{k}} = \sum_i e^{i\mathbf{k}\cdot\mathbf{r}_i}$ is the Fourier-space density. Orientational correlations are also considered, and are quantified using the six-fold bond-orientational order parameter [?][48]

$$\psi_6 = \frac{1}{N}\sum_{j=1}^{N}\psi_6^j \quad \text{where} \quad \psi_6^j = \frac{1}{n_j}\sum_{k=1}^{n_j}\exp(i6\theta_{jk}), \quad (3)$$

where the sum is performed over the $n_j$ first neighbors of the $j$-particle. These neighbors are defined as particles with $r_{ij}/\sigma_{ij} < 1.33$, which is the location of the distance of the first minimum in the rescaled radial distribution function $g(r/\sigma_{ij})$. The angle $\theta_{jk}$ then measures the orientation of the axis between the two particles with respect to the $x$-axis. Because these correlations are orientationally invariant the choice of $x$-axis is made without loss of generality. Orientational correlations are then monitored through the two-point bond-orientational correlation function

$$g_6(r) = \langle \psi_6(r)\psi_6^*(0) \rangle, \quad (4)$$

where $\psi_6(r) = \sum_{i=1}^{N}\delta(|\mathbf{r} - \mathbf{r}_i|)\psi_6^i$. The radial decay of the hexatic order correlation function, $g_6(r)/g(r)$[48], provides an hexatic correlation length $\xi_6$, as presented in Supplementary Note 1.

Translational dynamics is characterized by first measuring the intermediate scattering function

$$F_s(k, t) = \frac{1}{N}\left\langle \sum_{j=1}^{N}\exp\left[ik\cdot(r_j(t) - r_j(0))\right] \right\rangle \quad (5)$$

at the wave number $k$ corresponding to the first peak of $S(k)$. The relaxation time of the density fluctuations, $\tau_\alpha^{TR}$, is then extracted from the exponential decay of the scattering function, i.e., $F_s(k, \tau_\alpha^{TR}) = e^{-1}$. Orientational dynamics is characterized similarly, replacing the Fourier-space density by the bond-orientational correlation function in Eq. (3) defined by

$$C_{\psi_6}(t) = \frac{1}{N}\left\langle \sum_{i=0}^{N}\psi_6^i(t)\left[\psi_6^i(0)\right]^* \right\rangle. \quad (6)$$

In order to extract the bond-orientational relaxation time $\tau_\alpha$, we use $C_{\psi_6}(\tau_\alpha) = e^{-1}$.

**Equilibration and the glass ceiling**. Normal Monte-Carlo (MC) simulations allow only local particle displacements, drawing a random displacement vector on the $(x, y)$ axis in the interval $[-\Delta r_{max}, \Delta r_{max}]$ with $\Delta r_{max} = 0.6$ and moving a randomly chosen particle following a Metropolis acceptance criterion. Compounding $N$ such displacement attempts defines a MC step, which is used as unit of time in this work. To ensure equilibration, we monitor both static and dynamical observables. Starting from a high-temperature liquid configuration, we quench the system at the final temperature and wait for the potential energy of the system to stop aging on a time window of $\sim 10^6$ MC steps. We first estimate $\tau_\alpha$ on simulations long enough to allow few decorrelations of $C_{\psi_6}(t)$, and then perform simulations for $220\tau_\alpha$. The system is left to equilibrate during the first $20\tau_\alpha$; static and dynamical observables are computed over the following $200\tau_\alpha$. Swap MC simulations include attempts at exchanging random pairs of particle diameters, which replace particle displacements with probability $p_{swap} = 0.2$. This algorithm defines the SWAP dynamics. The same equilibration and measuring protocol as for normal MC is then followed. Static observables monitor ordering and phase separation in the system, as discussed in Supplementary Note 1, whereas dynamical observables quantify the relaxation and equilibration timescales.

In Supplementary Fig. 1, we report orientational $\tau_\alpha$ and translational $\tau_\alpha^{TR}$ relaxation times for both normal and SWAP dynamics. Because the relaxation of local orientational degrees of freedom is slower, the associated timescale is used as reference. We perform three different fits to the $\tau_\alpha$ results for the physical dynamics, in order to extract the temperatures relevant to the dynamical slowing down. First, we fit $\tau_\alpha$ to a power-law function, as is predicted in the context of the mode-coupling theory[49],

$$\tau_\alpha \propto (T - T_{MCT})^{-\gamma}, \quad (7)$$

over the interval $\tau_\alpha \in (\tau_0, 10^3\tau_0)$. The resulting $T_{MCT} = 0.123$ roughly corresponds to the lowest temperature at which normal dynamics can reach equilibrium in simulations of reasonable duration[24].

Next, we estimate the laboratory glass transition temperature, $T_g$, at which experiments with atomic and molecular glass formers cannot be equilibrated anymore. At $T_g$, relaxation times have increased by 12 orders of magnitude with respect to their value at the onset of the supercooled dynamics[50]. We thus fit the relaxation times both to a Vogel–Fulcher–Tallman (VFT) law

$$\tau_\alpha \propto \exp\left(\frac{A}{T - T^{VFT}}\right), \quad (8)$$

and to an Arrhenius law

$$\tau_\alpha \propto \exp\left(\frac{B}{T}\right), \quad (9)$$

where $A$ and $B$ are fitting constants. These two expressions respectively overestimate and underestimate the increase of relaxation times in experimental glass-formers[51,52]. We fit Eq. (8) using the whole temperature range $T < T_{onset}$, whereas we fit Eq. (9) only to $T < 0.16$ to ensure that the result serves as a proper lower bound on the relaxation time. Extrapolating up to the temperature at which $\log_{10}(\tau_\alpha/\tau_0) \simeq 12$ gives $T_g^{VFT} = 0.0907$ and $T_g^{Arr} = 0.0738$. These two temperatures are, by construction, upper and lower bounds for $T_g$, and thus define an experimental glass-ceiling regime (blue shaded region)[26] in Figs. 1, 2 and 3 as well as Supplementary Fig. 1. In all cases, SWAP dynamics equilibrates well beyond this experimentally limited regime, reaching $T = 0.026$. Supplementary Fig. 1 also shows the fitting curves to the dynamics. The mode-coupling power-law prediction describes the growth of the relaxation times only within the first three decades of the glassy regime, but at lower temperatures it overestimates the results by many orders of magnitude. Whereas Eq. (8) adequately describes these same results over more than four decades, an Arrhenius law captures barely two decades.

**Data availability**

The data necessary to reproduce the figures in this paper are publicly available through the Duke University Libraries Digital Repository (https://doi.org/10.7924/r46w9b248)[53].

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

## Acknowledgements

The authors thank G. Tarjus for stimulating discussions. This research was supported by a grant from the Simons Foundation (#454933, L. Berthier and #454937, P. Charbonneau). Part of the computations was carried out through the Duke Compute Cluster.

## Author contributions

L.B., P.C., A.N., M.O., and S.Y. designed research, performed research, analyzed data, and wrote the paper.

## Additional information

**Competing interests:** The authors declare no competing interests.

