## [Peer Review File · Nature Communications]

Reviewers' comments:

Reviewer #2 (Remarks to the Author):

First, I summarize my concerns relating to interpretations of the authors' simulations and the citation of results from other sources that should be considered in a revision:

The computations are restricted to estimating the configurational entropy S_c of a toy glass-forming liquid composed of polydisperse circular particles that enables their specialized Monte Carlo sampling method. Moreover, the computations are performed for a two-dimensional fluid for which well-defined transport properties do not exist in the thermodynamic limit. It is also a shortcoming that there is no generally agreed upon relationship between the configurational entropy and the structural relaxation time τ , so I do not understand how the authors can estimate T_g (normally defined by a temperature where the relaxation time in three dimensions is on the order of 100 s) or conclude that the effective relaxation time of their system is many times the lifetime of the universe. In fact, the authors cannot say anything about the dynamics of their liquid from S_c alone and the actual longest relaxation time determined by MD simulation is roughly on the order of 100 ns in physical units rather than 100 s. The authors have already published a paper on their Monte Carlo method of calculating the configurational entropy in polydisperse particles in three dimensions [ref. 18 of the paper] so that computational methodology itself is not novel.

On the other hand, there is circumstantial evidence supporting the Adam-Gibbs (AG) model linking τ to S_c and there is interest in testing this relationship, making the authors results of potential wide scientific interest in this connection. The evidence of the authors that their Monte Carlo estimate of S_c in two dimensions, at least in their toy liquid model, extrapolates to a value near zero is certainly suggestive that there is no finite temperature Kauzmann temperature in two dimensions, a finding emphasized in the title of the paper. This result is interesting and deserves publication, although I am not sure Nature Communications is the best place for publication because of the technical interest of this result. Upon rereading the paper, I began to appreciate Fig. 2. This rescaling procedure is very intriguing and offers a possible welcome reconciliation to the various incompatible estimates of the configurational entropy for the same fluid found in the literature based on different computational methods, a source of considerable confusion and controversy in the current scientific literature. It is mainly on the basis of Fig. 2 that I suggest publication of this paper in Nature Communications. Further discussion of this figure would be welcome.

Some general comments about issues that came up in the former review that can be avoided by modest revision:

I gather from reading the responses of the authors that that they are firm believers in the existence of a finite Kauzmann temperature at which S_c vanishes in three dimensions and that this critical condition is only prevented in experiments and simulations by the material systems going out of equilibrium. There is no proof of this intuition and experiment certainly does not support this expectation, as the authors imply. Rather, the measurements in three dimensions only show a common tendency for S_c (actually an excess entropy S_{exc} , since S_c cannot be experimentally determined and where it is often assumed heuristically that $S_{exc} \approx S_c$) to extrapolate to a zero at a finite temperature. This extrapolation procedure going back to Kauzmann is interesting, arguably useful from an engineering standpoint, but the actual vanishing of S_c at a finite temperature is an important *open scientific question*.

Gibbs and DiMarzio (GD) developed the first analytic theory suggesting that a vanishing of the configurational entropy for a glass-forming polymer liquid could be identified with a 'glass transition' and the authors have revised their paper to include this seminal work, which also focused attention on S_c . Subsequent theoretical work indicated that the mean field calculations of GD violated certain mathematical bounds and their conclusions were then brought into question, even though the Gibbs-DiMarzio has been found to predict numerous trends in the glass transition in diverse polymeric glass-forming liquids in the polymer science literature and the model is still widely utilized. Later simulation studies of the configurational entropy (density) [Wolfgang et al. PRE vol. 54, 1535 (1996)] and analytic calculations on a more sophisticated lattice cluster mean field theory polymer fluid thermodynamics developed by Freed and coworkers (Dudowicz et al. Adv. Chem. Phys. vol. 137, 125 (2008)) later clarified that the problem with original high GD theory was its high temperature expansion treatment of chain semi-flexibility, which invalidates the extrapolation of results of this model to too low temperatures. Further insight into this problem was obtained recently in another paper by Freed and coworkers [Xu et al. J. Chem. Phys. vol. 145, 234509 (2016)] which treats glass-formation in flexible polymer melts where the former high temperature expansion is not introduced. These new calculations show an initial steep decrease of $S_c(T)$ upon cooling, from which one could extrapolate a Kauzmann temperature, but below the estimated T_g (through a combination of the AG and lattice cluster thermodynamic models linking the relaxation time to S_c quantitatively), $S_c(T)$ turns over to approach a constant value, as reported in

numerous experiments (a point mentioned in my last review) and also found in Monte Carlo simulation estimates by Binder and coworkers [Wolfgang et al. PRE vol. 54, 1535 (1996)]. Note that this turnover is reported in Xu et al. and Betancourt et al. as an *equilibrium phenomenon* and the existence of this plateau brings into question the linear extrapolation of S_c to 0 based on data above the glass transition. Of course, all this discussion relates to $d=2$ so the authors may well be correct in their conclusion that S_c genuinely extrapolates to zero or becomes very small near $T=0$ in their model two-dimensional fluid.

Note that a saturation in S_c at low temperature has been found in statistical mechanical models of probable relevance to understanding glass-formation. Kohring et al. [PRL vol. 57, 1358 (1986)] found that the configurational entropy of the X-Y model in three dimensions (a model relevant to the superfluid and superconductivity transitions, and proposed to describe crystal melting) has a low temperature “ordered” state with a finite configurational entropy. Polymerization models involving equilibrium polymerization [Dudowicz et al. JCP vol. 111, 7116 (1999) and proposed as a model of cooperative particle cluster formation in glass-forming liquids [Douglas JCP vol. 125, 144907 (2006); Betancourt et al. JCP vol. 140, 204509(2014)] and demonstrated to describe equilibrium polymer formation in solution, gives rise to a finite residual entropy at low temperatures that scales inversely to the average mass of the dynamic associating polymeric species, as postulated by AG for their hypothetical cooperatively rearranging regions in their heuristic model of glass-formation. Recent simulation studies for many fluids have indicated the existence of dynamic clusters having a polymeric form exhibiting particle exchange motion whose degree of polymerization directly reflects the change of the activation energy for diffusion and structural relaxation. [Betancourt et al. JCP vol. 140, 204509(2014)] Note also that Feynman [Phys. Rev. vol. 90, 1116 (1953)]; *ibid.* vol. 92, 262 (1954)] formulated an attractive equilibrium polymerization model of the superfluid transition in He^4 , where he suggested that the atoms were undergoing permutational or “stirring” exchange motion that is similar to the string-like collective motion actually seen in simulations just mentioned in classical cooled liquids, and found both experimentally and in simulation for many other forms of strongly interacting condensed matter.

While it is fair to say that despite these observations and theoretical modeling efforts that no rigorous theory has yet emerged regarding glass-formation in three dimensions, it is clear that the emergence of a low temperature “glass state” having a finite configurational is certainly a possibility, at least in some liquids.

The approach of S_c to a low temperature plateau rather than vanishing has many practical implications for measurement. Within an AG model, or its string model extension [Betancourt et al. JCP vol. 140, 204509(2014)], the approach of S_c to a plateau at low temperatures means that relaxation becomes Arrhenius below the glass-transition temperature, which curiously occurs near the reported experimental glass transition T_g where the structural relaxation time equals 100 s. Measurements on glasses aged below the experimental T_g over millions of years to achieve equilibrium strongly support this predicted return to Arrhenius relaxation at low temperatures rather than a diverging relaxation time at a putative Kauzmann temperature. See also: O’Connell and McKenna, JCP vol. 110, 11054 (1999), Novikov and Sokolov, PRE vol. 92, 062304 (2015); Zhang et al. JCP vol. 145, 114502 (2016); Chung et al. JCP vol. 147, 154902(2017). There is also simulation evidence by Sciortino and coworkers of colloidal particles with grafted associating chains that form thermodynamic equilibrium glass-like state that appears very similar to strong glass-formers that exhibit Arrhenius relaxation under low T conditions [Soft Matter vol. 11, 3132 (2015); there are better references relating to by this group on this topic that I could not locate]

There is no convincing evidence for a diverging relaxation time in glass-forming liquids [Hecksher et al. Nature Physics vol. 4, 737 (2008)] as one would expect from a vanishing S_c . Ag inferred such a divergence in their glass transition model based on limited specific heat estimates of the excess entropy above T_g and extrapolation of the trend in this data to low temperatures where S_{ex} formally vanished, but the new work based on the AG model and the calculation of S_c for flexible polymers rather than the empirical rough experimental estimate of S_c indicates that S_c approaches a constant at low temperatures and that the relaxation corresponding becomes Arrhenius, as an equilibrium rather than a non-equilibrium phenomenon. [Xu et al. J. Chem. Phys. vol. 145, 234509 (2016)] It is clear that the AG model does not actually predict a divergent relaxation time at a Kauzmann temperature, as often stated in the literature. There is moreover no experimental evidence for large changes in the activation energy in real liquids that one might associate with the large scale collective motion that the authors suggest. The observed scale of collective motion in real liquids, as inferred indirectly from specific heat measurements or changes of activation energy from relaxation measurements, is generally smaller than about 10, usually much smaller. [Betancourt et al. JCP vol. 140, 204509(2014)]

As a point of terminology regarding the paper title. In the case of spin models, and other models exhibiting thermodynamic transition phenomena, it is usual terminology to say that no phase transition exists if the free energy has no singular variation in its free energy or derivatives occurs at a positive finite temperature. For example, the one-dimensional Ising model is normally said to have no phase transition in this sense, given the singular variation in this model occurs at zero temperature. By “glass transition” the authors seem to be instead to achieving the thermodynamic condition that $S_c=0$, which is not a condition for a phase transition (although Gibbs-DiMarzio identified this condition as such). It seems more correct in all senses of the term “transition” to say that the results of the authors suggest that *no glass transition exists in two dimensions*. In this connection, it seems worth noting previous work by Torquato [e.g., Donev et al. JCP vol. 127, 124509 (2007)] and others indicating that no ideal glass transition exists in binary hard discs. Notice that Torquato insert the word “ideal” to avoid confusion with the identification of “glass transition” with a “phase transition”. It is true that the authors make their terminology of what they mean by “glass transition” clear in their introduction, but I think the title has the potential for confusion because of different interpretations of what is meant by the term “glass transition temperature”

All these difficulties can be avoided if the authors leave dynamics out the discussion and simply focus on their results indicating an ideal glass transition in the sense of vanishing S_c as T approaching 0 in a model $d=2$ fluid.

Some comments in green are made to the authors' responses:

Referee 2

The referee cites many names and works without providing a single precise reference. It is thus not always easy to completely decipher what the referee had in mind.

We nonetheless did our best to interpret the statements offered, in order to provide the most precise and complete responses.

Referee's comment

While this paper discusses the results of an interesting Monte Carlo method for estimating the configurational entropy (s_c) of soft sphere particle mixtures, the authors make unwarranted claims based on their simulation data so that I think Nature is not

appropriate journal for publication.

I stand by the statements above, although the judgement is probably overly harsh because of my concerns regarding the interpretation of the authors' results with respect to dynamics.

Authors' response

First and foremost, we wish to highlight that the above summary properly reflects neither the content nor the message of our manuscript.

The referee erroneously assigns several "unwarranted claims" by other groups to our current work. The vast majority of the referee's comments deal with papers that report results or opinions that are not relevant for the data we present here.

We have provided a long supplementary file containing the necessary data and technical details and the list of complementary checks that establish our main results: (i) equilibration at 'cosmological' timescales; (ii) equilibrium measurements of configurational entropy lower than any previous work; (iii) careful comparison to modern theoretical work and available literature, when relevant. No such claims are 'unwarranted', and the referee fails to demonstrate that they are.

Our actual data are actually discussed in only two sentences of the report, and these brief comments even contain an obviously incorrect statement about crystallisation. In short, the referee does not address our results, but rather offers a general presentation of his or her personal beliefs and opinions.

We detail the assessment offered by this referee point by point below.

Referee's comment

Experiments do not indicate a vanishing of S_c below the glass transition, as ascribed to Kauzmann.

Authors' response

By construction, experiments report equilibrium measurements of the configurational entropy s_c down to T_g , at which point the system necessarily falls out of equilibrium by the definition of T_g . To conclude about the fate of the equilibrium s_c , one thus ought to rely on extrapolations from temperatures above T_g .

Conclusions based on such extrapolations are very dangerous, as the explicit calculation of S_c for flexible chains from the lattice cluster theory illustrates.

A naive extrapolation of these results suggest a vanishing of s_c at T_K , as has been heavily discussed in the literature since Kauzmann (e.g., J. Chem. Phys. 108, 9016 (1998) and Phys. Rev. Lett. 109, 045701 (2012) - Refs. [13,14] of the manuscript, respectively).

Yes, naïve is an appropriate phrasing.

Of course, when experiments go out of equilibrium below T_g , they do report a non-equilibrium saturation of the configurational entropy, but this is different from what we do. Here, we provide equilibrium measurements to extremely low temperatures (that would completely bypass the experimental glass temperature T_g), and we equilibrate the system down to $T_g/3$, which no experiment can achieve.

I really don't see how the authors can define T_g in two dimensions if they do not perform low temperature MD simulations or cannot even relate S_c to the structural relaxation by any quantitative relation.

Doing so, we confirm the absence of any discernible saturation of the equilibrium configurational entropy below T_g , provided the system remains equilibrated. Our work is, therefore, not only in

agreement with experiments (in the sense you use the same type of extrapolation assumption), but it extends the available regime of equilibrium measurements of the configurational entropy (This admittedly seems to be true and interesting). In our resubmitted manuscript, we carefully mention that all measurements are performed in the equilibrated fluid, and never in the non-equilibrium glass. Although this may well be true, it is hard to actually prove you are in equilibrium.

Referee's comment

Measurements point to a positive residual configurational entropy in the low temperature glass state (as discussed by Johari) above $T = 0$ so that the Kauzmann paradox of a vanishing S_c seems to be more a matter of an improper extrapolation than a real physical phenomenon.

I have amplified on this point above. A finite residual configurational entropy at low temperatures is a real possibility that is in line with many relaxation observations in materials in the glass state after long aging times that are required for equilibration. Of course, the "effective" value of S_c will drift over time at low temperatures towards its equilibrium value as the system slowly comes into equilibrium so non-equilibrium effects and aging effects are prevalent in real glassy materials.

Authors' response

The referee probably refers to a long discussion of Johari with Kivelson Reiss, Gupta, Mauro, and Goldstein, about the reality of the residual entropy (e.g., J. Phys. Chem. B, 103, 8337 (1999), J. Chem. Phys., 128, 154510 (2008), J. Non-Cryst. Solids, 355, 595 (2009)). As one can see directly in Johari's papers (e.g., J. Chem. Phys. 134, 034515 (2011)), the residual entropy is argued to emerge over the course of irreversible processes at T_g . In other words, the residual entropy is an out-of-equilibrium phenomenon that does not relate to the Kauzmann paradox, which is itself concerned with the fate of equilibrium fluids. Said differently, the residual entropy considered by Johari is an out-of-equilibrium effect that concerns low-temperature glasses, and therefore does not have any direct bearing on the equilibrium liquid properties measured in our work. Hence, the above line of work does not in any way interfere with our fully equilibrated data, because at no point in our manuscript do we consider low-temperature, out-of-equilibrium glasses. Hence, we do not discuss these works, nor do we contradict any of the arguments provided in those discussions. It is true that equilibrium measurements therefore require some extrapolation. Our central point is that this extrapolation can be performed more comfortably for us than anyone else before because we reach unprecedentedly low temperatures in equilibrium, and also that this extrapolation suggests a zero-temperature ideal glass transition. so even if 'improper', the trend offered by our extrapolation is new and exciting.

Referee's comment

The vanishing of S_c was first inferred in the mean field theory of Gibbs and DiMarzio where the existence of an ideal glass transition was identified with the vanishing of s_c at a finite positive T and was moreover associated with a second order phase transition. Unfortunately, this work was later shown to violate rigorous free energy bounds by Gudrati and Nagle. Although this type of mean field result was long ago discredited, the idea of s_c being critically small near T_g has continued to have some currency in the field of glass-formation (How can the authors not mention Gibbs-DiMarzio model of glass formation, which has had an even greater impact as the original phenomenological arguments of Kauzmann?)

Authors' response

The referee is correct that our manuscript should cite the work of Gibbs and DiMarzio (J. Chem. Phys. 23, 3 (1958)), because the paper offered the first theoretical interpretation of Kauzmann's observations. It is an oversight from our part that we have now corrected. In our resubmitted manuscript, we have added a sentence citing Gibbs and DiMarzio. However we do not pay attention to the details of their mean-field simple lattice polymer model, which is considered inadequate to describe the polymer glass transition, as discussed by Nagle (J. Phys. Chem. 88, 4599 (1984)). The subsequent re-examination of this model by Gudrati and Goldstein (J. Phys. A: Math. Gen. 13, L437 (1980), J. Chem. Phys., 74, 2596 (1981), and J. Stat. Phys. 28, 441 (1982)) therefore does not present much concern for our work. We mention in passing that the referee's remark reects an impressive lack of knowledge of the current theoretical literature.

I am aware, but not impressed by the "current theoretical literature" indicated.

Since the Gibbs-DiMarzio work 1958, a proper mean-field theory of the glass transition has been derived (and recently reviewed in Annu. Rev. Condens. Matter Phys. 8, 265 (2017) { Ref. [6] of our manuscript), and this derivation becomes mathematically exact in the limit of large spatial dimensions. (This is not the case of the Gibbs-DiMarzio 'mean-field' model, which is simply a coarse-grained, approximate, theoretical treatment.) In this mathematically well-defined limit, the vanishing of a properly-defined configurational entropy is a mathematical result, that can, obviously, not violate any bound.

As explained above, the Gibbs-DiMarzio model, a model of real glass-forming liquid and a mathematically correct definition of the configurational entropy. The problem with this model is that it involves a high temperature expansion, a point clarified by subsequent work by Freed and coworkers where this "old" theory is found to correspond to only a leading term in a systematic cluster expansion of the thermodynamic properties of polymer fluids. This lattice cluster theory is well validated by a large body of thermodynamic data and should not be dismissed so readily. Do the authors really think that their model is anything but a highly "coarse-grained" model of any fluid existing in the physical world.

Therefore, the referee's comment about an old idea having some 'currency' demonstrates ignorance of modern theoretical methods to describe glass formation. What does remain unknown from these advances is whether the vanishing entropy predicted in the mean-field limit can survive in finite dimensions (especially in two dimensions, as in our manuscript), a question that none of the literature quoted by the referee succeeds to answer.

This is indeed the main question and the authors' work provides some interesting evidence that it might be possible for S_c to vanish in a toy liquid, albeit not at a finite temperature. Above I note that there is at least one type of cooled liquid that does not exhibit a Kauzmann condition at finite temperature in $d=3$, but even this result doesn't provide that it cannot happen for another kind of fluid, as Stillinger has noted before.

In our resubmitted paper, we have reformulated the content and degree of rigor of the mean-field theory that we use as theoretical background and motivation to our numerical measurements.

Referee's comment

Recent calculations by Freed and coworkers, based on an extended lattice model of molecular uids with bending energies, as well as enthalpic interactions and molecular

bending energy, indicate that s_c indeed approaches a residual value for $T > 0$ for $d > 2$. Notice this trend is apparent in ref. 13 of the authors, although non-equilibrium effects probably influence the estimate of the low temperature s_c in this paper.

Authors' response

As the referee correctly states, Freed and coworkers recently calculated polymer glass formation with varying spacial dimensions, based on their hypercubic lattice model with finite bending energy (Adv. Chem. Phys., 161, 443 (2016)). However, contrary to what the referee states, this computation suggests that the configurational entropy s_c does vanish for $2 < d < d_c$ (where d_c is near 8) and that above d_c , a positive residual entropy appears. (The referee might have reached an erroneous conclusion by misreading the scale of the vertical axis in Fig. 1 of that paper.) Yet, as the authors of that paper discussed, their lower-dimensional results might involve unphysical artifacts due to their use of a high-temperature expansion.

This is a concern in the results of Freed et al., as they admit, but this problem does not arise in the treatment of flexible polymers where the high temperature expansion treatment of chain flexibility is not required. The authors have no molecular bonds in their artificial polydisperse sphere fluid so they have they do not have to contend with the serious problems that arise from treating the effect of bending rigidity on glass-formation- a fundamental aspect of real molecular glass-forming liquids not even under discussion by the model described by the authors.

In any case, the low-dimensional and low-temperature results of that paper, which represent an approximation in any finite dimension, cannot be used to criticize our numerical results, which are direct and quantitative measurements.

The problem is not with the results themselves, which I find interesting, but with their interpretation.

In addition, the bending of s_c in Ref. [14] ([13] in the previous manuscript) (Phys. Rev. Lett. 109, 045701 (2012)) is there again a pure non-equilibrium effect, and thus irrelevant for the present study, which is exclusively concerned with equilibrium states, as already stated in our comments above.

This is a reasonable possibility.

Referee's comment

Measurements by McKenna and coworkers, Sokolov and coworkers and Doremus all provide consistent evidence that the VFT relation no longer applies below T_g , but rather returns to Arrhenius relaxation in glassy regime, consistent with experiments, simulations (Binder and coworkers) and measurements all indicating a finite residual entropy in the glass state below, T_g .

The point here is that there is evidence in three dimensional systems that S_c approaches an equilibrium finite residual value (or at least this can happen in some fluids), given sufficient time for the fluid to equilibrate. This physical phenomenon has nothing to do directly with the authors' simulations in $d=2$ where the situation could be quite different for many reasons.

Authors' response

Regarding the dynamics, our work does not use, prove, disprove, address, confirm, the validity of the Vogel-Fulcher-Tammann (VFT) law discussed in the experimental works mentioned by the referee. It may be true (or not) that the VFT law breaks down in some systems below the experimental T_g in three dimensions.

But this question has absolutely no bearing on any of our thermodynamic measurement, which do not indicate any qualitative change (for our two-dimensional glass

former) across the experimental glass transition temperature regime all the way down to $T_g=3$. Notice that to estimate timescales at very low temperatures, we do use an Arrhenius τ consistent with the above experiments to avoid overestimating the relaxation time with, for instance, the VFT law. We have otherwise already addressed above that the out-of-equilibrium residual entropy of the glass does not concern us here.

In conclusion, our thermodynamic data, which require no assumption and no hypothesis (such as the VFT law), indicate a steep decrease of the configurational entropy in $d = 2$ glass formers that contradicts absolutely no equilibrium experimental results that we know of, and certainly not those mentioned by Referee 2.

In our resubmitted manuscript, we now briefly describe the dynamical counterpart to our measurements to avoid the ambiguity putatively suggested by this comment. In particular, we have added the following paragraph:

The problem of the glass transition has two fundamental facets: thermodynamics and dynamics. While the current study focused on the thermodynamics of $d = 2$ glass formers, its dynamical counterpart, which involves obtaining a detailed functional form of the structural relaxation time, remains for now out of reach of computational work. Our results nonetheless provide strong constraints on the divergence of that relaxation time. In $d = 2$, in particular, any such divergence must take place at zero temperature.

Well, this is fine! Absolutely! The authors have a valid story in $d=2$, along as the results are appropriately qualified.

Referee's comment

As another point, uid transport properties such as mass diffusion coefficient and shear viscosity do not exist in $d=2$ dimensions so the interpretation of the glass transition in $d = 2$ dimensions is problematic. How do we know that the ratio of relaxation times indicated in the paper should apply to a two- dimensional fluid? How do the results change with system size? What is the relation between the relaxation time τ from the intermediate scattering function and τ_c for a two-dimensional system having a particular system size?

Authors' response

These questions all and a precise quantitative answer in the submitted supplementary information (SI). For the sake of this response, we summarize again our findings and methods here. As the referee correctly states, transport coefficients are affected by Mermin-Wagner (MW) long-range fluctuations in $d = 2$. This effect has been thoroughly examined, especially over the last few years (e.g., Proc. Natl. Acad. Sci. U. S. A. 114,101850 (2017) and Proc. Natl. Acad. Sci. U. S. A. 114, 1856 (2017) - Ref. [20,21] of our manuscript, respectively). We here build on these advances by using the state-of-the-art techniques developed in these works. The contribution of MW fluctuations are removed from both our dynamical (Section III of the SI) and our thermodynamic (Section V of the SI) measurements, as we detail below.

There is still the problem of defining meaningful transport properties in two dimensions. Stuart Rice and others have discussed this problem at length and he would be a good person for the editors to consult on this matter. He might be a good alternative reviewer if the editors would like another opinion on the merits of the present work since he is an acknowledged expert on quasi-two dimensional liquids and their theoretical idealizations in terms of two dimensional fluid models.

> How do we know that the ratio of relaxation times indicated in the paper should apply to a two-dimensional uid? How do the results change with system size?

Density correlation functions are affected by MW fluctuation, and hence the associated relaxation time exhibits strong finite-size effects. By contrast, the orientational dynamics is not affected by MW fluctuation, and thus provides a proper definition of a relaxation time in $d = 2$ (Nat. Commun. 6, 7392 (2015) - Ref. [19] of our manuscript). It is precisely this scheme that we employ in our work. The absence of significant finite-size effects on the orientational dynamics is further confirmed in FIG. 1 of the SI.

Yes, they depend weakly on finite size, but these effects never really go away. Again the expertise of Stuart Rice would be useful to consult on this point.

>What is the relation between the relaxation time τ from the intermediate scattering function and sc for a two-dimensional system having a particular system size?

Whereas we can access equilibrium configurations at extremely low temperatures by means of the swap MC, we cannot obtain the associated relaxation time of the standard MC dynamics for these configurations when temperature is too low. Hence, a graph of the (orientational dynamics) relaxation time vs. sc cannot be generated in that regime. In any case, although interesting in its own right, the relationship between τ and sc is far beyond the scope of the current work.

Referee's comment

Recent simulations by Sastry and coworkers have found deviations in applying the Adam-Gibbs model relation in $d = 2$ in soft particle mixtures (Kob-Anderson model). I have the same problem with Sastry's work in $d=2$ as with the present paper and I have reminded him of this problem for some time. I also have great difficulty with Chandler (deceased), Garrahan, and others when they speak of the "breakdown" of the Stokes-Einstein relation in $d=2$. How can this relation break down if it doesn't exist in the first place in $d=2$ because of the Stokes paradox? The problem with defining diffusion coefficient and viscosity in $d=2$ in the thermodynamic limit is a consequence of the recurrent nature of random walks in two dimensions. The vortex excitations associated with momentum diffusion have to diffuse to infinity to have dissipation. [Douglas Comp. Mat. Sci. vol. 4, 292 (1995)]

Authors' response

Sastry and coworkers have indeed reported that the Adam-Gibbs relation (connecting thermodynamics and dynamics) is violated in $d = 2$, whereas the relationship seemingly holds in $d = 3$ and 4 (Phys. Rev. Lett. 109, 095705, (2012) - Ref. [22] of our manuscript). Notice that these observations refer to the temperature regime corresponding to the first few decades of glassy slowdown, rather than the very low temperature regime in which we here measure the configurational entropy. (Notice that Sastry et al. did not remove the MW fluctuations from the relaxation dynamics discussed above by the referee which might explain their results.) But as mentioned already above, we do not perform dynamical measurements at very low temperatures, and we do not assess, use, prove or disprove the validity of the Adam-Gibbs relation. However, the observation by Sastry and coworkers suggests a certain peculiarity of $d = 2$ glasses that might be related to the absence of the finite-temperature TK we report here. This explains why we cited that work in our initial submission.

The mathematics is interesting, but the relevance to practical applications is a question. This brings us back again to relevance of publishing this work in Nature Communications.

Referee's comment

One additional problem, is that the Kob-Anderson model crystallizes so it is entirely unclear that soft sphere models such as the one investigated in the present paper are suitable for studying molecular glass-forming liquids. In short, I have a lot of problems with the model considered in this paper.

The referee is correct to highlight the importance of avoiding crystallization and other ordering types when studying model glass formers, such as the Kob-Anderson (not \Anderson") model. In fact, Figs. 1(a) and (b) of the main text are dedicated to this issue by means of both real and reciprocal space inspections. We have carefully checked for the absence of crystal (FIG. 1(a) of the main text) and hexatic (FIG. 1(b) of the main text) ordering in the system we study. Ample additional demonstration that our model does not crystallize is also provided in Section VI of the SI. Finding glass forming models that strongly resist crystallisation was actually one the obstacles faced in developing the SWAP algorithm, as amply discussed in the earlier publication "Models and algorithms for the next generation of glass transition studies" (Phys. Rev. X 7, 021039 (2017) cited as Ref. [17] of our manuscript).

In short, there is no "problem" with our model and methodology, as all information is provided in the main and SI manuscripts, and our main methods have been validated already in earlier published works.

The last sentence by the referee is a clear statement about a superficial reading of our submission (which includes the SI in case of "problems with the model"), and a cursory knowledge of both recent theoretical and numerical literature.

Referee's comment

No phase transition occurs at a finite temperature, even within the authors extrapolation.

Authors' response

Yes, this is exactly our point, as stated in the title of the manuscript.

Your point would be clearer if you stated that there is no "ideal glass transition" to avoid confusion with phase transition. Other authors have suggested that glass formation corresponds to a kind of phase transition so the authors leads to the potential for confusion.

Referee's comment

Comparisons between thermal glass-formers and ultra-stable glasses are not suitable.

I still think equating stable and unstable glasses are like equating apples and oranges and should be avoided. The vapor deposition are often in often in a far from equilibrium state, while thermal glass-formers may at least approach an equilibrium condition after a long aging time.

Authors' response

Computer glasses produced SWAP cover a range of stability that seems to defy the referee's imagination. In earlier work, we have shown that these systems can correspond { in terms of stability { to systems prepared even below the experimental glass transition, very much in the spirit of ultrastable glasses. For instance, we have performed direct quantitative measurement of kinetic stability, following a protocol devised in the group of M. Ediger who discovered ultrastable glasses and found competitive results. This comparison has even been quantitatively established in a previous paper (EPL, 119 36003 (2017)).

In a recent review article (J. Chem. Phys. 147, 210901 (2017) cited as Ref. [3] in our manuscript), M. Ediger himself compares his ultrastable systems to results performed using SWAP, and he has publicly expressed that he is engaged in a "friendly competition" with the SWAP results. Other groups have produced very stable systems already, and have fully validated our comparison to ultrastable glasses.

That statement, therefore, simply expresses the referee's lack of understanding of the performance, limitations, and success of the numerical method we use in our work that effectively produces ultrastable glasses in three dimensions, and even better glasses in two dimensions, as we report here.

Referee's comment

Torquato and Stillinger have recently suggested that fluctuations give rise to a residual entropy in the glass state, an argument that seems rather likely to me.

Authors' response

The configurational entropy measurements by Torquato and Stillinger (J. Chem. Phys. 127, 124509 (2007)) display a residual entropy due to the same non-equilibrium effects as Ref. [14] (Phys. Rev. Lett. 109, 045701 (2012)) mentioned above. As one can see from FIG. 13 of that paper, T_g and the associated residual entropy systematically depend on the cooling rate (compression rate in their case), which is a hallmark of an out-of-equilibrium phenomenon. In addition they argue (with arguments that are not "rather likely" to be correct, in our view) that this residual entropy is bounded by the mixing entropy of the glass. (Notice that this bound is totally ill-defined for our polydisperse model, and therefore "rather unlikely" to hold here). Some of us have addressed the question of the mixing entropy in two recent works (J. Chem. Phys. 146, 014502 (2017) and J. Chem. Phys. 149, 154501 (2018) - Refs. [24] and [26], respectively), in which it is demonstrated that this bound is due to an improper treatment of the mixing entropy. Again, this remark shows a lack of knowledge of the modern theoretical literature on some issues, and confuses a body of work dedicated to the nonequilibrium glass to our equilibrium measurements.

Yes, estimates of configurational entropy are often observed to drift with cooling rate or after long aging times, but this does not negate the possibility that these systems can eventually come into equilibrium so that a true equilibrium configurational entropy in the glass state can exist. There is significant circumstantial experimental evidence that this situation occurs in some real glass-forming liquids. No general proof one way or the other exists so I think we should be open-minded on the issue.

Referee's comment

Glass-formation ($d > 2$) in the form of a phase transition reminiscent of a second order phase transition might indeed in athermal systems, and there is recent simulation data supporting this interesting possibility for such jammed materials.

Authors' response

The jamming transition is observed for packings of particles in absence of thermal fluctuations. Jamming can also be studied with hard disks in the limit of infinite pressure obtained by strong compression of the nonequilibrium hard sphere glass.

These studies differ conceptually from the measurements we perform for models in the presence of thermal fluctuations, and that are at thermal equilibrium. The only connection to jamming we see is that for the hard disk results reported in the SI, which suggest that the Kauzmann transition might in fact coincide with jamming for this

system (infinite pressure for hard disks being equivalent to zero temperature for soft disks, in this context). Yet the qualitative features of the configurational entropy of this system are not controlled by any of the concepts devised in recent years to understand the jamming transition itself. We notice in addition that within the mean-field theory of glasses that we mention in the introduction (that the referee seems to be unaware, see above),

I am aware of the theoretical results mentioned. The thing I find most interesting about this new mean model is that it refutes the validity of the formerly “rigorous” mean field model of glasses- mode coupling theory. It is good to see that the community of glass theorists, many of which formerly rejected any thermodynamic basis for glass formation, have now embraced the importance of a thermodynamic transitional phenomena and the central significance of the configurational entropy.

This indicates great progress, in my view, I strongly support the efforts of the authors to calculate S_c accurately to help build a sound theory of both the thermodynamics and dynamics of glass-forming liquids. This brings back to appreciating the highly positive aspects of this work that I neglected in my first review.

both glass and jamming transitions can be analytically addressed in a unified way in the large dimensional limit, see e.g. *Annu. Rev. Condens. Matter Phys.* 8, 265 (2017) (Ref. [6]), but theory also clear shows how different these transitions are at the conceptual level.

My point here was related to the role of thermal fluctuations on the dynamics of glass-forming liquids based on comparing the phenomenology of thermal glasses to their driven athermal counterparts. This matter is of admittedly secondary relevance to this paper, but I was stimulated by their title to think about the possibility of a true phase transition like behavior in glass-forming systems. I can see now that the authors really did not mean to invoke the notion of phase transition by their term “glass transition” so this former comment is not relevant to a review this paper. Sorry for my confusion on this matter.

Referee's comment

I am sure the authors could alter the paper readily to focus on the novelty of their results for a model $d=2$ fluid and the philosophical interest of these results and thus have a nice publication. It is not clear, however, that such a publication would be suitable for *Nature Communications* since it not clear how their results relate to the dynamics of real glass-forming liquids. This is a question the editors to decide.

In summary, I thus do not understand, nor believe, that the authors are simulating any physical system having a relaxation time 10^{18} times the lifetime of the universe. The molecular simulations are rather just the usual relaxation times accessible in this type of simulation method- nothing special. The precise meaning of these statements is discussed at the beginning of my long rereview of this paper. I now admit that these statements, while technically supportable, were overly harsh and neglected the positive and novel aspects of the paper.

Authors' response

For all reasons mentioned above, this statement is simply outrageous. The referee is not asked to believe our results, but indeed to understand them.

Our manuscript reports a detailed series of demonstrations and quantitative tests and measurements and a carefully documented SI, building on several published lines of evidence as well as independent conformations by several other groups that the SWAP algorithm produces glasses with a stability far beyond what can be experimentally pro-

duced. Stating that our simulations are “nothing special”, which presumably means “no different from ordinary molecular dynamics simulations” is thus objectively incorrect. Closing the report with such an erroneous statement about our work with no scientific justification is unacceptable.

The referee is also supposed to judge the measurements (numerical data) themselves, and to correlate them with the modern theoretical state of the art. Finally the referee should also be able to compare our results with the relevant experimental literature. We believe that the report by Referee 2 fails to accomplish any of these criteria.

Reviewer #3 (Remarks to the Author):

Using their SWAP Monte Carlo algorithm (e.g., refs. 17, 18), the authors demonstrate the existence of a thermodynamic glass transition in a two-dimensional polydisperse soft sphere system at a temperature numerically indistinguishable from 0. The transition corresponds to the vanishing of the configurational entropy (Kauzmann entropy crisis.)

This is an important result, which demonstrates the existence of a thermodynamic (ideal) glass transition in a finite-dimensional system. Key to the reliability of this calculation is the SWAP algorithm's ability to extend the range of temperatures over which the configurational entropy can be computed down to approximately 1/3 of the glass transition temperature.

The technical quality of the work is very good, and the paper is very clearly written. The authors' responses to the referees (especially referee 2) are clear and correct. Several of this referee's comments reflect an insufficient knowledge of recent theoretical developments in glass physics, as well as a misunderstanding of the manuscript's scope (e.g., comment about the validity of the VFT equation, when in fact the authors do not address dynamics.)

I recommend publication of this interesting and important work in Nature Communications.

Pablo G. Debenedetti

Reviewer #4 (Remarks to the Author):

The manuscript by Bethier et al is a sound and enjoyable piece of work carrying interesting new information about the glass transition.

The main claim of this paper is that the new Monte Carlo algorithm, named SWAP, allows to examine a 2D model of glass former at temperature where equilibrium studies were previously numerically inaccessible (by far) and which is also out of the reach of experimental observations.

The authors take advantage of this powerful technique, and of other methods well consolidated beforehand, to unveil the physics hidden in this unexplored range of temperature and in particular to shed light on the long standing open question about the existence of a glass thermodynamic transition in finite dimensions, two dimensions in this case.

The result is therefore definitely of interest to the broad audience of Nature Communications.

I am undoubtedly in favour of publication of the present manuscript on this journal after a few minor improvements are implemented.

Previous iterations with other referees have largely focused on objections about the main coordinates of the work, to which the authors have given extensive and motivated answers.

However not much has been considered about the actual content and its presentation especially regarding the discussion of the results and the conclusions, which in my opinion can be sharpened a jot for the sake of clarity.

To this aim, please find below a list of issues that might require some rethinking/reformulation.

pg1 first column

concerning the efforts to include "finite d fluctuations beyond the mean field framework" it could be worth mentioning the recent attempt in

<https://journals.aps.org/prb/abstract/10.1103/PhysRevB.98.174205>

<https://journals.aps.org/prb/abstract/10.1103/PhysRevB.98.174206>
to highlight the fact that it is a pretty active field nowadays

pg1 end of second column

Instead of "This remarkable advance reveals the existence of a thermodynamic glass transition at $T_K = 0$ for $d = 2$, accompanied by an entropy crisis and the divergence of the point-to-set correlation length." I would suggest to write "This remarkable advance give new sturdy evidences in favour the existence of a thermodynamic glass transition at $T_K = 0$ for $d = 2$, accompanied by an entropy crisis and the divergence of the point-to-set correlation length." or something similar along those lines. Indeed, extrapolations, even if in a smaller range of temperature, are still required to assess the existence of such a transition, so doubts can still be cast about whether the glass transition occurs at $T=0$ or some finite little entropy is still left at $T=0$ or finally other mechanisms out from the observed range of temperature could intervene to avoid the transition itself.

pg1 end of second column

"Our results thus illuminate the dimensionality dependence of the glass transition and shed light on the nature of glassy dynamics in $d = 2$ [19,21]." to be possibly changed into "Our results thus shed light on the low dimensional fate of the glass transition and on the nature of glassy dynamics in $d = 2$ [19,21]." as only the $D=2$ case is studied and nothing new is discussed about the "dimensionality dependence" of the glass transition, which still stands as a wide open problem.

pg2 second column

I would not suggest to say that "the SWAP equilibration algorithm bypasses the slowdown associated with the glass transition in $d = 2$." rather than "the SWAP equilibration algorithm partially bypasses the slowdown associated with the glass transition in $d = 2$." or " rather that "the SWAP equilibration algorithm significantly speeds up the glass dynamics in $d = 2$." or any other reformulation that go in a similar direction. Indeed, some amount of slowing down is still present and an eventual divergence of the unphysical SWAP relaxation time is expected (as the authors are of course well aware: see arXiv:1805.12378).

fig2(b) and the corresponding discussion in the main text (pg3 beginning of first column)

fig2(b) should show a linear vanishing of $s_{\text{conf}}=a \cdot T - T_K$. This linear behaviour arguably has some quadratic correction at larger T therefore it is proposed a fit with a quadratic function.

It would be useful to have the result of the fitting parameters either in the main text (for example the one corresponding to the curve show in the figure) or in the SI (I could not find this information there either) so to immediately see the small T_K appearing as a result and the relative weight of the linear term and the quadratic correction.

Moreover the corresponding discussion in the SI (section SCALING) about the different fitting for s_{conf} is rather confusing. All the different forms proposed can simply be motivated by the typical expected behaviour for s_{conf} . This behaviour has a linear trend when it vanishes, and a bending at higher temperatures that motivate the quadratic correction. This is equivalent to the other proposed form for $1/s_{\text{conf}}$ as $1/T + B$, since $1/s_{\text{conf}}$ is not expected to vanish when T gets larger, or at least not in a continuous way. The correspondence of the two fitting forms is therefore not a fact that is only checked a posteriori as proposed in the SI, rather both functional forms naturally stem from the same qualitative behaviour. I suggests to motivate such functional forms when proposed and show their obvious correspondence as a corollary, rather than highlighting an a posteriori "consistency" as if it were fortuitous.

Finally the scaling of ξ with T is a mere consequence of the linear behaviour of s_{conf} at small T and of the fact that ξ and s_{conf} have exactly the same trend as it was already shown in the main text. However, it could be interesting to note that the pre-critic quadratic correction, that is easily visible in

the plot s_{conf} vs T , does not correspond to any visible deviation from the linear behaviour when the plot shows $1/s_{\text{conf}}$ vs $1/T$ or ξ_{PS} vs $1/T$, suggesting that pre-critic corrections are not that problematic for the study of the critical exponent of ξ_{PS} .

fig3(a)

I could not find any explanation of the blue dashed curve in the figure, it is the same set of data as fig 2(b), as one can guess from the following sentence only "For any $T > 0$, this crossover around $R \sim \xi$ " so the curve should be the same as the one reported in fig2(b) but I believe that a clearer explanation is needed either in the main text or at least in the figure's caption.

pg4 first column

I easily see the connection of the phase diagram T vs $1/R$ with the one obtained from the random pinning approach, less so the connection with the result of the epsilon coupling procedure, which originates a first order transition line rather than a glass transition. In any case for the random pinning phase diagram, among the many possible citations I would strongly suggest to cite the theoretical paper(s) which contain the full discussion about the transition and the phase diagram

<https://www.pnas.org/content/109/23/8850> (and may be

<https://aip.scitation.org/doi/10.1063/1.4790400>). (Note that in case the authors also want to cite numerical studies of the random pinning phase diagram one cannot avoid to mention the more recent <https://www.pnas.org/content/112/22/6914>.)

Also, <https://onlinelibrary.wiley.com/doi/pdf/10.1002/9781118202470.ch2> would represent a useful reference to the reader in reference with the discussion briefly sketched in "For any $T > 0$, this crossover around $R \sim \xi_{\text{PTS}}$ corresponds to a finite-size version of the random first-order glass transition with a rarefaction of the number of locally available states as R decreases", about the comparison between point to set and random first order transition. Note that the theoretical papers <https://www.pnas.org/content/109/23/8850> and <https://aip.scitation.org/doi/10.1063/1.4790400> give for the random pinning glass transition exactly the same explanation, except from the finite-size bit, as the one proposed here, hence the parallel between the two phase diagrams. So here it is another reason of the importance to not to forget to mention these papers in such a context.

Finally the following reference to a first order transition might sound misleading as it is: "The evolution of $P(Q)$ in Fig. 3(b) indeed exhibits features reminiscent of phase coexistence near an incipient first-order transition. The crossover also..." since first order transitions are not associated to the divergence of correlation time as the glass transition. I suggest to be more cautious and avoiding comparisons that cannot be circumstantiated by writing "The resulting crossover is expected to manifest into a rapid change of the order parameter, as it is confirmed by the evolution of the $P(Q)$ in Fig. 3(b). The crossover also..." or something similar.

pg4 end of first column

One should be careful in stating that

"These results are consistent with the sharp decay of the configurational entropy in Fig. 2 and the dramatic increase of the relaxation time in Fig. 1." because fig1 does not show physical relaxation time in the range of temperature for which "large clusters comprising about 140 particles are statically correlated". A more careful discussion about correlation time follows, therefore I simply suggest not mention the relaxation time here.

pg4 beginning of second column

The following sentence is not clear: "Our results nonetheless provide strong constraints on the divergence of that relaxation time." what are these constraints? The conclusion of the paper are still based on extrapolations, I do not see any strong constraints on the divergence emerging from the results or their discussion.

pg4 beginning of second column

Be more circumstantiated on the statement "In $d = 2$, in particular, any such divergence must take place at zero temperature." so that it would be intelligible to a broad audience such as the one of Nature Communications

pg4 second column

"show that a thermodynamic transition can occur in finite-dimensional systems" in my opinion should be replaced by the less-conclusive "give strengthened evidences that a thermodynamic transition can occur in finite-dimensional systems" or similar rewording.

Reply

Referee # 2

The report of Referee 2 is now clearly much more positive than at the previous stage. We are happy to read that referee 2 better appreciates the several novel aspects of our work.

It is clear that the referee is an expert of polymer glass theories, which provides an interesting and constructive perspective on the glass problem. But it also means that a large part of the discussion in the report remains specific to polymeric systems that are in some way very different from the class of model systems we simulate in this work. In addition, even the most refined theoretical developments discussed for polymeric systems do not quite suffice to obtain direct insight into our results. We nevertheless take the opportunity provided by such results to include the discussion of possible alternative theoretical scenarios in the revised manuscript. Also, while Referee 2 correctly emphasizes that no experimentally decisive measurement exist regarding either the existence of a finite temperature Kauzmann transition or the divergence of a relaxation time, it is important to highlight that there does not exist any definite experimental proof against them either. We believe we have kept an “open mind” on these issues since our first submission.

Because the second report by referee 2 is quite lengthy and contains several layers of exchanges, we summarize below the key points that need a response, and offer a reply. We also explain for each point the related changes to the manuscript.

Generality of our results

The referee mentions that we cannot easily extend our results obtained for two-dimensional glass forming liquids to three-dimensional systems, and also to distinct types of materials such as polymers and patchy colloids. In particular, the referee argues at length that theoretical proposals stemming from polymer science that S_{conf} could saturate as $T \rightarrow 0$ to a finite value in equilibrium conditions.

Our views are that: (i) as the referee admits, the existence of a finite residual entropy for polymers is a theoretically-valid idea; (ii) for point particles there is no strong theoretical support for this idea, and most recent theoretical treatments suggest a

different scenario; (iii) our simulations are not directly relevant for the study of polymers or patchy colloids. Because off-lattice point particles with isotropic interaction constitute a markedly simpler glass model, is the only one for which an exact theoretical treatment exists in the large dimensional limit, and has been used over the last 30 years in thousands of computer simulations, we consider its study to nonetheless be of acute and general interest.

In practice, we have modified the introduction to include the mention of more complex models (such as polymers and patchy colloids) citing relevant articles (mainly by the Freed and Sciortino groups). We also mention recent theoretical developments in these areas (in particular the series of works by the Freed group listed by the referee). In the presentation of the numerical model, we again justify why the model we study is good for our purposes, and why the results we obtain are of general interest both from theoretical AND experimental viewpoints. Finally, we acknowledge the possibility that a saturation of the entropy can occur at a yet inaccessible low temperature in our model.

Existence of transport coefficients in two dimensions

The referee points out an issue about transport coefficients in two dimensions. It is indeed known that time correlation functions of liquids at modest temperature/density show power-law decay, the so-called long-time tail, which leads transport coefficients, such as the diffusion constant, to diverge in two dimensions. The referee therefore doubts existence of the transport coefficient and relaxation time in our case.

This issue has been widely discussed and clarified over the last 20 years in the field of computational studies of glass-forming liquids.

In particular, it has been reported both experimentally and computationally that long-time tails are no longer detected in low-temperature or very dense supercooled liquids. See some experimental papers, Vivek et al., Proc. Natl. Acad. Sci. U. S. A. 114, 1850 (2017), Illing et al., Proc. Natl. Acad. Sci. U. S. A. 114, 1856 (2017), and a recent computer simulations paper, Flenner and Szamel arXiv:1808.10004 (2018), further validate this point. The long-time tails themselves therefore become buried under the slow glassy relaxation in the regimes relevant to the study of supercooled liquids.

Moreover, it has been reported that orientational correlation functions (that we here use to extract a relaxation time) show an exponential decay even at modest density hard disks, as shown in Isobe and Alder J. Chem. Phys. 137, 194501 (2012), and that long-time tails play strictly no role for such functions.

All the relaxation times reported in our paper are therefore insensitive to issues related to long-time tails, and meaningful relaxation times can be used to describe the slow dynamics of our two-dimensional systems.

Because this issue is well-known and discussed at length in many earlier studies that we already cite, we have not significantly changed the text of the manuscript. We simply state that we can measure the dynamic relaxation in our system in two dimensions unambiguously, without going again through the full scientific reasoning. We have nonetheless added a sentence stating that the dynamical observables used are not affected by long-time tails, and that the relaxation times considered are all well defined.

How to estimate T_g in our simulations

The referee misunderstands how we estimate the experimental glass transition temperature T_g in our simulations, and wrongly believes that we have used the hypothesis from the Adams-Gibbs relation between entropy and dynamics.

We have addressed this point in our first reply already, and the definition of timescales and the determination of T_g are described at length in Section III of the SI, so we here only briefly summarize how we proceed once again.

We measure the relaxation time (with no ambiguity as mentioned above) from the trajectory of the Monte-Carlo (MC) simulations over the accessible range. It is well known that MC dynamics is akin to Brownian dynamics in so far as the glassy regime is concerned (Berthier and Kob, *J. Phys.: Condens. Matter* 19, 205130 (2007)). We then fit the temperature dependence of these data using a range of several fitting functions (two of them do not include any divergence, and one is the Arrhenius law) to extrapolate and extend the dynamical regime by some orders of magnitude. We then determine T_g when the relaxation time has grown by 12 orders of magnitude beyond the onset temperature. The uncertainty in the extrapolation from different fitting functions is shown in the plots. In short, the dynamics is carefully measured, the extrapolations are well controlled, and the Adam-Gibbs relation is not used.

The title of this paper

In short, and after careful reconsideration of this issue, we decided to stand by the current title.

Our results and analysis do indicate that the configurational entropy and the point-

to-set correlation length support a critical behaviour at $T = 0$, with no sign of any saturation down to temperatures that are about $T_g/3$. This temperature is much lower than any turnover temperature of S_{conf} seen in any of the theories mentioned by the referee. We have not a single numerical argument to support any other view, for now, even though something could conceivably still happen at a yet inaccessible temperature.

We now mention such possibility in the body of the text, but keep our extant title because referee's proposal does not do justice to our results.

Comparison with ultrastable glasses

It is correct that ultrastable glasses prepared far below T_g are likely far from equilibrium. But the Ediger group has demonstrated that slightly below T_g , vapor deposited glasses fall in the extrapolation of the equilibrium system and thus correspond to nearly equilibrium glasses prepared below T_g . Our comparison with ultrastable glasses stands in the sense that we also obtain equilibrium configurations below T_g .

Note that we do not mention “unstable” glasses, but rather “ultrastable” glasses in order to highlight that we obtain states well below T_g . This terminology simply suggests that our results below T_g are relevant and insightful for such experimental systems. We do not make any claim that appears unwarranted.

Referee # 3

We thank referee 3 who offers a very positive report, and offers direct support regarding our earlier reply to Referee 2.

No specific questions were raised by the referee.

Referee #4

We thank referee 4 for the very positive report and the recommendation to publish. We also thank the support offered regarding previous correspondence.

The referee also makes some constructive remarks that we now address.

-Referee's comment: *pg1 first column concerning the efforts to include "finite d fluctuations beyond the mean field framework" it could be worth mentioning the recent attempt in <https://journals.aps.org/prb/abstract/10.1103/PhysRevB.98.174205> <https://journals.aps.org/prb/abstract/10.1103/PhysRevB.98.174206> to highlight the fact that it is a pretty active field nowadays*

-We have added citations to those papers and to two others on the same topic.

-Referee's comment: *pg1 end of second column Instead of "This remarkable advance reveals the existence of a thermodynamic glass transition at $TK = 0$ for $d = 2$, accompanied by an entropy crisis and the divergence of the point-to-set correlation length." I would suggest to write "This remarkable advance give new sturdy evidences in favour the existence of a thermodynamic glass transition at $TK = 0$ for $d = 2$, accompanied by an entropy crisis and the divergence of the point-to-set correlation length." or something similar along those lines. Indeed, extrapolarions, even if in a smaller range of temperature, are still required to assess the existence of such a transition, so doubts can still be cast about whether the glass transition occurs at $T=0$ or some finite little entropy is still left at $T=0$ or finally other mechanisms out from the observed range of temperature could intervene to avoid the transition itself.*

-We agree and have now rephrased that our work "gives very strong evidence".

-Referee's comment: *pg1 end of second column "Our results thus illuminate the dimensionality dependence of the glass transition and shed light on the nature of glassy dynamics in $d = 2$ [19,21]." to be possibly changed into "Our results thus shed light on the low dimensional fate of the glass transition and on the nature of glassy dynamics in $d = 2$ [19,21]." as only the $D=2$ case is studied and nothing new is discussed about the "dimensionality dependence" of the glass transition, which still stands as a wide open problem.*

-We agree and have rephrased into “Our results thus illuminate the low-dimensional fate of the glass transition”

-Referee’s comment: *pg2 second column I would not suggest to say that “the SWAP equilibration algorithm bypasses the slowdown associated with the glass transition in $d = 2$.” rather than “the SWAP equilibration algorithm partially bypasses the slowdown associated with the glass transition in $d = 2$.” or “rather that “the SWAP equilibration algorithm significantly speeds up the glass dynamics in $d = 2$.” or any other reformulation that go in a similar direction. Indeed, some amount of slowing down is still present and an eventual divergence of the unphysical SWAP relaxation time is expected (as the authors are of course well aware: see arXiv:1805.12378).*

-We agree that the SWAP itself eventually slows down. We have thus changed the sentence to state that the method “largely bypasses” the dynamical slowdown.

-Referee’s comment: *fig2(b) and the corresponding discussion in the main text (pg3 beginning of first column) fig2(b) should show a linear vanishing of $s_{\text{conf}} = a(T - T_K)$. This linear behaviour arguably has some quadratic correction at larger T therefore it is proposed a fit with a quadratic function. It would be useful to have the result of the fitting parameters either in the main text (for example the one corresponding to the curve show in the figure) or in the SI (I could not find this information there either) so to immediately see the small T_K appearing as a result and the relative weight of the linear term and the quadratic correction.*

-We have added the values of the fitting parameters both into the text and into the SI.

-Referee’s comment: *All the different forms proposed can simply be motivated by the typical expected behaviour for s_{conf} . This behaviour has a linear trend when it vanishes, and a bending at higher temperatures that motivate the quadratic correction. This is equivalent to the other proposed form for $1/s_{\text{conf}}$ as $1/T + B$, since $1/s_{\text{conf}}$ is not expected to vanish when T gets larger, or at least not in a continuous way. The correspondence of the two fitting forms is therefore not a fact that is only checked a posteriori as proposed in the SI, rather both functional forms naturally stem from the same qualitative behaviour. I suggests to motivate such functional forms when proposed and show*

their obvious correspondence as a corollary, rather than highlighting an a posteriori "consistency" as if it were fortuitous.

-We agree. We motivate the quadratic fit shown in the main text. Then, in the SI we present the second fitting as being the corollary of that offered in the main text, as this is indeed more logical.

-Referee's comment: *Finally the scaling of ξ with T is a mere consequence of the linear behaviour of s_{conf} at small T and of the fact that ξ and s_{conf} have exactly the same trend as it was already shown in the main text. However, it could be interesting to note that the pre-critic quadratic correction, that is easily visible in the plot s_{conf} vs T , does not correspond to any visible deviation from the linear behaviour when the plot shows $1/s_{\text{conf}}$ vs $1/T$ or ξ_{pts} vs $1/T$, suggesting that pre-critic corrections are not that problematic for the study of the critical exponent of ξ_{pts} .*

-When we plot $\xi_{\text{pts}} = A(1/T) + B$, there is no visible deviation from the linear scaling. The critical exponent is therefore robustly estimated. One can clearly observe that the pre-critical correction in the s_{conf} vs. T plot (Fig. 2(b)) is due to the presence of the offset B . We now mention this issue in the SI.

-Referee's comment: *fig3(a) I could not find any explanation of the blue dashed curve in the figure, it is the same set of data as fig 2(b), as one can guess from the following sentence only "For any $T > 0$, this crossover around $R \sim \xi$ " so the curve should be the same as the one reported in fig2(b) but I believe that a clearer explanation is needed either in the main text or at least in the figure's caption.*

-The dashed blue line was indeed not described. The caption now explains that the curve is the same fit that was discussed in the last two comments.

-Refere's comment: *pg4 first column I easily see the connection of the phase diagram T vs $1/R$ with the one obtained from the random pinning approach, less so the connection with the result of the epsilon coupling procedure, which originates a first order transition line rather than a glass transition. In any case for the random pinning phase diagram, among the many possible citations I would strongly suggest to cite the theoretical paper(s) which contain the full discussion about the transition and the phase diagram <https://www.pnas.org/content/109/23/8850> (and may*

be <https://aip.scitation.org/doi/10.1063/1.4790400>). (Note that in case the authors also want to cite numerical studies of the random pinning phase diagram one cannot avoid to mention the more recent <https://www.pnas.org/content/112/22/6914>.) Also, <https://onlinelibrary.wiley.com/doi/pdf/10.1002/9781118202470.ch2> would represent a useful reference to the reader in reference with the discussion briefly sketched in "For any $T \gtrsim 0$, this crossover around $R = \xi_{\text{pts}}$ corresponds to a finite-size version of the random first-order glass transition with a rarefaction of the number of locally available states as R decreases", about the comparison between point to set and random first order transition. Note that the theoretical papers <https://www.pnas.org/content/109/23/8850> and <https://aip.scitation.org/doi/10.1063/1.4790400> give for the random pinning glass transition exactly the same explanation, except from the finite-size bit, as the one proposed here, hence the parallel between the two phase diagrams. So here it is another reason of the importance to not to forget to mention these papers in such a context.

-We have added some of the suggested theoretical references. They should indeed help readers better understand the issue at stake.

-Referee's comment: *Finally the following reference to a first order transition might sound misleading as it is: "The evolution of $P(Q)$ in Fig. 3(b) indeed exhibits features reminiscent of phase coexistence near an incipient first-order transition. The crossover also..." since first order transitions are not associated to the divergence of correlation time as the glass transition. I suggest to be more cautious and avoiding comparisons that cannot be circumstantiated by writing "The resulting crossover is expected to manifest into a rapid change of the order parameter, as it is confirmed by the evolution of the $P(Q)$ in Fig. 3(b). The crossover also..." or something similar.*

-We have rephrased this passage to avoid any possible confusion with conventional first-order transitions.

-Referee's comment: *pg4 end of first column One should be careful in stating that "These results are consistent with the sharp decay of the configurational entropy in Fig. 2 and the dramatic increase of the relaxation time in Fig. 1." because fig1 does not show physical relaxation time in the range of temperature for which "large clusters comprising about 140 particles are statically correlated". A more careful discussion about correlation time follows, therefore I simply suggest not mention the relaxation time here.*

-We have rephrased the part related to the relaxation because we indeed do not observe the dynamics directly over the entire range, although dynamics certainly does slow down as temperature drops further below where we can measure it.

-Referee's comment: *pg4 beginning of second column The following sentence is not clear: "Our results nonetheless provide strong constraints on the divergence of that relaxation time." what are these constraints? The conclusion of the paper are still based on extrapolations, I do not see any strong constraints on the divergence emerging from the results or their discussion. pg4 beginning of second column Be more circumstantiated on the statement "In $d = 2$, in particular, any such divergence must take place at zero temperature." so that it would be intelligible to a broad audience such as the one of Nature Communications*

-We have rephrased this passage as: "Our results nonetheless suggest that in $d = 2$ the divergence of the relaxation time must take place at zero temperature."

-Referee's comment: *pg4 second column "show that a thermodynamic transition can occur in finite-dimensional systems" in my opinion should be replaced by the less-conclusive "give strengthened evidences that a thermodynamic transition can occur in finite-dimensional systems" or similar rewording.*

-We have toned down that sentence accordingly.

REVIEWERS' COMMENTS:

Reviewer #4 (Remarks to the Author):

As far as I can see the authors have discussed and largely addressed all the issues raised. I am therefore in favour of publication.